# Microsatellite Analysis Revealed Potential DNA Markers for Gestation Length and Sub-Population Diversity in Kari Sheep

**DOI:** 10.3390/ani12233292

**Published:** 2022-11-25

**Authors:** Muhammad Ibrahim, Sohail Ahmad, Israr Ud Din, Waqas Ahmad, Ijaz Ahmad, Sher Hayat Khan, Ihtesham Ul Haq, Jehan Zeb, Olivier Andre Sparagano

**Affiliations:** 1Institute of Biotechnology and Genetic Engineering, The University of Agriculture Peshawar, Peshawar 25130, Pakistan; 2College of Veterinary Sciences, The University of Agriculture Peshawar, Peshawar 25130, Pakistan; 3Department of Infectious Diseases and Public Health, City University of Hong Kong, Hong Kong SAR, China; 4Department of Zoology, Abdul Wali Khan University Mardan, Mardan 24200, Pakistan

**Keywords:** Kari sheep, gestation length, microsatellite association, bottleneck analysis, genetic diversity

## Abstract

**Simple Summary:**

Livestock production needs to be improved to ensure global food security and meet the growing demand for food. A sheep’s gestation length (the period from conception to birth) is usually about 150 days. We have discovered a novel sheep breed in Chitral, Pakistan, whose individuals may gestate for a relatively shorter period, i.e., 100 days, and yet can give birth to normal lambs. To explore the underlying genetic causes for this unique character we used DNA markers. This approach helped us identify some contrasting genetic variations among short-gestating (Kari-S) and long-gestating (Kari-L) Kari ewes. Two of the total 31 targeted DNA sites demonstrated an association with gestation length in Kari sheep. Following an in-depth analysis and further confirmation in other sheep breeds, these DNA sites may be linked to genes controlling gestating length in sheep. The findings from this study may open ways for faster sheep reproduction with reduced cost.

**Abstract:**

Kari sheep inhabiting the Chitral district of Pakistan show variation in gestation length. In this study, we have analyzed the genetic differences between the three subtypes of Kari sheep (based on variation in gestation length) using microsatellite markers. Kari sheep samples were collected from their breeding tract and were characterized for gestation length and genetic diversity using microsatellite markers. A total of 78 Kari ewes were grouped into three categories based on gestation length (GL), i.e., Kari-S (with a shorter GL), Kari-M (with a medium GL), and Kari-L (with a longer GL). DNA from these samples was used to amplify 31 ovine-specific microsatellite loci through PCR. Of the total 78 Kari specimens, 24 were grouped in Kari-S (GL = 100.7 ± 1.8), 26 were from the Kari-M subtype (GL = 123.1 ± 1.0), and 28 were Kari-L (GL = 143.8 ± 1.5). Microsatellite analysis revealed an association of genotypes at two marker sites (MAF214 and ILSTS5) with variation in GL. A total of 158 alleles were detected across the 22 polymorphic loci with an average of 7.18 alleles per locus. Unique alleles were found in all three subtypes. The highest number of unique alleles was observed in Kari-L (15), followed by Kari-S (10) and Kari-M (8). The results indicated that Kari-S is a genetically distinct subtype (with higher genetic differentiation and distance) from Kari-M and Kari-L. The genetic uniqueness of Kari-S is important for further exploration of the genetic basis for shorter gestation length, and exploitation of their unique values.

## 1. Introduction

Kari is a small-sized, thin-tailed sheep breed found in the Chitral district (The North West of Khyber Pakhtunkhwa province) of Pakistan [1]. The area comprises the rugged mountains of the Hindukush and Karakorum ranges. Nature has endowed Chitral with 35 narrow and deep valleys. Most of the area is rangeland with scattered vegetation (62%), glaciers and snow (24%), and forest (4%); about 3% of the land is cultivable [2].

The population of Kari sheep is limited in number (probably less than 15,000); however, it is in high demand by the residents for their social and cultural festivities. The consumption of mutton increases by 22% at the household level during the winter, which is very harsh and prolonged in the region [3]. Apart from mutton production, Kari is renowned for the fiber (wool) it produces, which is used by the local cottage industry for making Patti (a woolen cap) and other hand-made woolen fabrics [4].

The gestation length of most of the sheep breeds ranges between 140 and 145 days with little variation reported [5]. Gestation length in sheep is affected by different environmental, physiological, and genetic factors. For example, changes in photoperiod during gestation have been shown to extend gestation length by 2.4 days and lower the birth weight by 0.5 kg [6]. Gestation also gets longer by a reduction in feed and nutrient intake by the dam during pregnancy [7]. A direct association has been found between the age of ewes (explicated from the length of the telomeres in white blood cells) and gestation efficiency in Soay sheep [8]. Variations in gestation length among individuals within the same population have been reported in other animal species [9,10]. While in other species the gestation length variation may be more prominent, little variations are reported in sheep, usually only a few days.

Kari sheep have a mean gestation length of 110 days and a litter size of 1–2, yielding 3.88 lambs/annum. Due to the shorter gestation length, Kari has received much attention from the local farmers; however, crossbreeding is in practice to enhance wool production and adult weight. It is posing a risk to the uniqueness and genetic pool of Kari sheep, which still needs to be discovered [1].

DNA markers are in use to detect genetic diversity and connectivity among populations [11], for species identification [12], to identify paternity [13], and for their association with production or disease traits [14,15]. The use of microsatellite markers has been applied to identify alleles associated with fertility (number of lambs born) and birth weight in Merino sheep [16]. Microsatellite-based identification of quantitative trait loci for gestation length in sheep could make the selection process more accurate which will result in faster genetic improvement. Furthermore, it is essential to measure the genetic diversity of domestic livestock genetic resources as a prerequisite to the application of conservation strategies.

Kari sheep is a unique genetic resource reported to possess large variations in gestation length among individuals [17]. In this study, we first characterized and grouped the experimental animals belonging to Kari sheep breeds based on the variation in gestation length. Next, we undertook an association analysis of the different ovine-specific microsatellite markers with the gestation length variability. Finally, data from the microsatellite DNA markers were used by applying different tools to assess overall genetic diversity among and within the identified Kari sheep subtypes.

## 2. Materials and Methods

### 2.1. Animal Selection and Experimental Design

Kari is a unique sheep heritage showing large variation in its gestation length ranging from 87 to 153 days and thus, in some cases, this period lasts only for three months. For this reason, the breed has been divided into three categories: (i) animals having short (87–95 days) gestation length; (ii) animals having medium (120–123 days) gestation length; and (iii) animals having long (140–153 days) gestation length [17]. This variation in gestation length is prominent in the different breeding seasons and geographic locality of the sheep.

Flocks consisting of purebred Kari sheep specimens were surveyed in the Chitral district. Sheep having different gestation as reported by the shepherds were identified and purchased. A total of 78 ewes and three rams were acquired and transferred to a control shed near Chitral city. Before acquiring, the selected ewe specimens were confirmed open (no pregnancy) through ultrasound examination. The animals were grazing in mountain rangeland as well as stall-fed with concentrates. Ewes and rams were kept segregated at all times and were only allowed to mate as required. After acclimatization for a week, the sheep’s estrous cycle was synchronized using 0.25 mg/mL cloprostenol and subsequent doses of 60 mg/ewe medroxyprogesterone acetate. The rams were allowed to the flock after 36 h of the final treatment with medroxyprogesterone acetate injection. The conceptions were confirmed by ultrasonic observation after two months of the mattings. The flock was allowed to gestate and lamb normally. Data were recorded for gestation length, litter size, and birth weight for each lambing. Blood samples (3 mL per sheep) were collected from all 78 ewes in pre-labeled vacutainers and transferred on ice to the laboratory for DNA extraction.

### 2.2. Isolation of Genomic DNA from Blood

Genomic DNA was isolated from whole blood samples using a standard phenol-chloroform extraction process along with DNA extraction buffer containing SDS (20%), NaCl (400 mM), Tris base (100 mM), EDTA (100 mM) and proteinase K (1 mg/mL).

### 2.3. Polymerase Chain Reaction

The polymerase chain reactions were conducted in 15 μL reaction volume. Each reaction tube comprised 100 ng genomic DNA, 0.25 μM primer (both forward and reverse), 200 μM dNTPs, 1× Taq buffer including KCl, 1.5 mM MgCl2, and 1 unit of Taq DNA polymerase. The thermal profile was as: an initial denaturation at 94 °C, followed by 30 cycles with each cycle consisting of a denaturation step of one minute at 94 °C, an annealing step of one minute at primer-specific annealing temperature (Appendix A), and an extension step of 2 min at 72 °C. The last cycle was followed by a 10 min extension step at 72 °C. GeneAmp PCR System 2700 (Applied Biosystem, Waltham, MA, USA) programmable thermocycler was used for the amplification reactions. A total number of 31 ovine-specific microsatellite markers (Appendix A) were used in the amplification reaction with every individual DNA sample. The PCR products obtained after amplification were analyzed on 10% polyacrylamide gel along with a 50 bp DNA ladder.

### 2.4. Data Collection

The banding pattern formed by the PCR products on polyacrylamide gels was scored manually, and the size of each band was calculated by plotting its position on the standard curve using a 50 bp DNA ladder. The ladder provided sufficient gradation to identify small differences (up to three nucleotides) in band sizes (Figure 1). For each primer and sheep individual, a genotype was assigned based on the band size, homozygosity (giving a single band), and heterozygosity (giving a double band). The number of alleles for each microsatellite marker was counted by the differences in sizes of the PCR products on the polyacrylamide gels. The data were further analyzed using statistical software (see below).

### 2.5. Statistical Analysis

Means and standard errors for gestation length, litter size, and birth weight were calculated for the three Kari subtypes, i.e., Kari-S, Kari-M, and Kari-L categorized as being based on the variation in gestation length. One-way ANOVA along with PostHoc Tukey’s test was applied using SPSS version 23.0 to evaluate significant differences among the three groups. Principal component analysis (PCA) was performed in SPSS using rotation varimax at 25 iteration convergence to identify loci associated with GL variation in Kari sheep.

Variations in the banding pattern of microsatellite markers among the three subtypes of Kari sheep were analyzed using different computer software packages. For each subtype, the observed and effective number of alleles, observed and Nei’s unbiased expected heterozygosities, and allele frequencies were computed using POPGENE software version 1.32 [18]. A Chi-square test was performed to calculate probability values for Hardy-Weinberg equilibrium (HWE) at each locus within each of the three subtypes. The *p*-values for HWE at all the loci were then averaged to get the mean HWE *p*-value for each subtype. Formula by Botstein et al. [19] was used for the calculation of PIC values from the allele frequency data for every locus, in each subtype. The PIC values of all the polymorphic loci within each population were then averaged to get a mean PIC value for each subtype. Frequencies of null alleles were calculated at each locus using GENEPOP software version 4.3 [20]. The Ewens-Watterson test to check the neutrality of the microsatellite loci in each of the three subtypes was performed at 10,000 simulated samples using POPGENE.

The proportion of individuals correctly assigned to their respective population of origin was done by GENECLASS software [21] using different statistical approaches, i.e., Bayesian methods proposed by Rannala and Mountain, and Baudouin and Lebrun; frequency-based method; and distance-based methods, i.e., Nei’s standard distance, Nei’s minimum distance, Nei’s average distance, Cavalli-Sforza cord distance, and shared allele distance.

To quantify the inbreeding estimates and population substructure, Wright’s fixation indices: FIS (an indication of intra-population genetic divergence), FIT (between populations inbreeding), and FST (inter-population genetic differentiation), were calculated using FSTAT Version 2.9.3.2 [22]. Allelic richness (the measure of allele number with equal frequencies corrected for sample size) was also calculated using FSTAT. Probability values for population inbreeding (FIT) and differentiation (FST) were calculated using the exact G-test in FSTAT based on 1000 randomizations, assuming random mating within the samples. Nei’s unbiased genetic distances and gene flow (NM) between combinations of the three subtypes were also calculated using POPGENE. A dendrogram was constructed based on Nei’s genetic distance using the unweighted pair group method with arithmetic mean (UPGMA).

To sort each individual into different subtypes, structure analysis was done using STRUCTURE software version 2.3 [23]. The program was run at a burnin length of 50,000 followed by 50,000 simulation repeats, using an admixture model of ancestry. The allele frequency model was set at correlated allele frequency. The dataset was evaluated using the K values (number of given ancestral populations) 2 and 3. The results of the three simulations run on STRUCTURE were uploaded to Clumpak (http://clumpak.tau.ac.il/; accessed on 10 November 2022) to obtain an average graph at the identified K values.

The allele frequency data were used in the BOTTLENECK software version 1.2.02 [24] to test the mutation drift equilibrium (whether the effective population size of each subtype has been maintained or reduced in the recent past) in the populations. The software computes this effect by taking the allele number and frequency data, at polymorphic loci, as input. Four statistical tests were performed in BOTTLENECK software namely, sign test, standardized differences test, Wilcoxon sign rank test, and a qualitative test of mode shift. Each of the first three tests calculates loci with heterozygosity excess under the stepwise mutation model (SMM), infinite allele model (IAM), and two-phase model (TPM). The mode shift test distributes the number of alleles observed in high to low-frequency classes.

## 3. Results

### 3.1. Gestation Length Variation among Kari Subtypes

A 100% conception and lambing rate were achieved in the flock. Based on gestation length (GL) the Kari ewes were grouped into three subtypes, i.e., ewes exhibiting shorter GL (Kari-S), ewes having a medium GL (Kari-M), and ewes showing longer GL (Kari-L). Of the total 78 Kari ewes acquired for this study, 24 ewes belonged to the Kari-S subtype showing shorter GL ranging from 87 to 112 days (Table 1). Medium GL (114–130 days) was recorded in 26 ewes grouped in the Kari-M subtype. The remaining 28 ewes exhibited longer GL (132–155 days) representing the Kari-L subtype. All the lambs were normal and alive at the time of birth. No effect of the variation in gestation length was observed on litter size and birth weight among the three subtypes.

### 3.2. Association of Microsatellite Polymorphism and Gestation Length

PCA revealed associations among microsatellite loci and GL variation in Kari sheep. Of the total 22 loci, 8 showed a positive correlation with GL, while the remaining 14 loci were negatively correlated. Among the positively correlated loci, MAF214 and ILSTS5 showed the strongest correlation with correlation values of 0.456 and 0.412, respectively, and plotted in the same cluster with GL on the PCA plot (Figure 2A). Analysis of mean GL among genotypes of highly associated loci revealed significant differences (Figure 2B). For marker MAF214, GL was significantly shorter in genotypes AF and BB compared to the genotypes AE and CE (with a dominance of allele E). Similarly, ILSTS5 genotypes BF and CF showed shorter GL compared to the genotypes containing alleles D and G (i.e., AD, DF, DG, and FG). For sizes and frequencies of these alleles refer to Appendix A.

### 3.3. Genetic Diversity among the Subtypes

#### 3.3.1. Intra-Subtype Genetic Variation

The microsatellite markers used showed genotype variations between the three Kari subtypes. Of the total 31 markers used, nine did not amplify successfully in any of the Kari subtypes. The remaining 22 markers were found polymorphic, giving collectively 158 alleles overall loci in all subtypes. One marker (OarFCB193) in Kari-S and one (BM8125) in the Kari-L subtype were monomorphic. The number of polymorphic loci and the total number of alleles generated by them in each of the three subtypes of Kari sheep are given in Table 2. The number of alleles for polymorphic loci ranged from 2 (OarFCB193) to 12 (OarJMP58 and INRA63) overall subtypes (Appendix A). The parameters of genetic variability for each population are presented in Table 2.

Kari-S showed a lower mean observed number of alleles (Na) compared to Kari-M and Kari-L subtypes. In individual subtypes the Na ranged from 2 (SRCRSP1 and HUJ616) to 8 (OarVH72) in Kari-S; 2 (MAF33 and OarFCB193) to 10 (INRA63) in Kari-M; and 2 (MAF65 and OarFCB193) to 9 (OarVH72, DYMS1, OarJMP58, and INRA63) in Kari-L subtype. The mean effective number of alleles (Ne) was lower than Na in all subtypes. Allelic richness (AR) was similar across the three subtypes of Kari sheep (Table 2).

In all three subtypes of Kari sheep the observed heterozygosity (Ho) was higher than expected (He). Within population inbreeding estimates (FIS) were smaller and were not significant in all three subtypes. In Kari-S the value for FIS was negative. Mean polymorphism information content (PIC) was higher in all subtypes. Within individual populations, The PIC value was less than 0.25 for one locus (OarFCB193) in Kari-L. Averaging the *p* values of HWE at the polymorphic loci in each population revealed that overall the populations were in HWE. However, within individual populations, five loci deviated from HWE in Kari-S, 14 loci deviated in Kari-M and 12 loci deviated in Kari-L.

#### 3.3.2. Allelic Polymorphisms

The three subtypes of Kari sheep shared a considerable number of alleles (Appendix A). Of the total 158 alleles observed at 22 microsatellite markers, only 54 were shared by all the three Kari subtypes. Kari-M and Kari-L shared a higher number of alleles compared to the allele shared by any of them with Kari-S. Unique alleles were also found in the three Kari subtypes at different loci. In total, 10 unique alleles were observed in Kari-S, eight in Kari-M, and 15 in the Kari-L subtype. Of the total unique alleles, five in Kari-S, two in Kari-M, and five in Kari-L were relatively more frequent (frequencies exceeded 0.2) than the others (Appendix A). Among the markers, ILSTS11 being the most polymorphic showed a higher number of unique alleles; four unique alleles were found at this locus, one in Kari-S, one in Kari-M, and two in the Kari-L subtype.

Alleles at different microsatellite markers, their sizes, and respective frequencies in the three Kari sheep subtypes are presented in Appendix A. Variation was observed in the frequencies of different alleles among Kari subtypes. The same allele at a higher frequency was observed at only eight out of 22 microsatellite markers in all three subtypes. Variable allele frequencies were observed at other loci among different subtypes with some alleles dominant in one subtype at a given locus. Null alleles were observed at more loci in Kari-L (14 loci), followed by Kari-M (12 loci), and Kari-S (7 loci).

#### 3.3.3. Ewens-Watterson Test for Microsatellites’ Neutrality

The Ewens-Watterson test was performed on the microsatellite data to detect whether the subtypes were the result of selection in the Kari population. Most of the markers were found neutral for selection in all three subtypes of Kari sheep, having observed F-value within the 95% confidence intervals (Figure 3). However, two makers (INRA63 and BM1329) in Kari-S and three markers (OarFCB193, BM1824, and SRCRSP1) in Kari-L subtypes fall out of the limits of the confidence interval. Markers MAF214 and ILSTS5 that showed association with gestation length were found neutral for selection in all the three Kari subtypes.

### 3.4. Genetic Assignment of Individuals to Their Respective Subtype

The percentage of individuals correctly assigned to their respective subtype obtained by putting the individual genotype data in GENECLASS software is presented in Table 3. All the individuals were correctly assigned to their subtype at a moderate probability. Sample specimens from Kari-L subtypes were assigned at relatively high precision: ranging from 58 to 83% under different statistical models. Assigning accuracy of Kari-S and Kari-M populations ranged from 50 to 83% and 54 to 73%, respectively.

### 3.5. Population Sub-Structure

To estimate population differences, the fixation indices FIT (Total inbreeding estimates) and FST (a measurement of differentiation) along with NM (gene flow) among different subtypes of Kari sheep were calculated using the FSTAT program (Table 4). The mean FST values were significantly higher among pairs of the three Kari subtypes, having *p* < 0.05 under the exacttest. Significant inbreeding (FIT) was recorded for Kari-M with Kari-S and Kari-L. However, the FIT was non-significant between the Kari-S and Kari-L subtypes. Similarly, higher gene flow (NM) was recorded among the Kari-M and Kari-L subtypes, while the Kari-S subtype showed relatively smaller NM with Kari-M and Kari-L. Estimates for genetic distances calculated among the three Kari subtypes revealed that Kari-S was the most distant of the three subtypes (Table 4). Kari-M and Kari-L showed less genetic distance in between compared to the genetic distance between either of them with Kari-S. The dendrogram (Figure 4) obtained from Nei’s unbiased genetic distance using the neighbor-joining method also showed that Kari-M and Kari-L subtypes were closest to each other.

The STRUCTURE analysis grouped different subtypes in inferred clusters (at the given K value) based on the allele frequency data at each locus (Figure 5). At K = 2, the Kari-S individuals were grouped in a separate cluster, while Kari-M and Kari-L shared one cluster with some admixture from Kari-S. This distinct clustering of Kari-S persisted at K = 3. Kari-M and Kari-L also made distinct clusters at K = 3; however, the clusters made by these two Kari subtypes showed an admixed ancestry. Only 75 and 59% of individuals from the Kari-M and Kari-L subtypes, respectively, clustered into separate inferred groups of the two populations.

### 3.6. Mutation Drift Equilibrium in Kari Subtypes

The results of the BOTTLENECK analysis showed that there were significantly higher numbers of loci with heterozygosity excess than expected under all mutation models in Kari-S and Kari-L subtypes with higher T2 values (Table 5). However, the Kari-M subtype showed a statistically similar number of loci with heterozygosity excess as expected (*p* > 0.1 under the SMM model) and with lower T2 value under standardized differences test. The results of the mode-shift test from the BOTTLENECK analysis, distributing the allele frequency data in different frequency classes are graphically presented in Figure 5.

The allele frequencies of the Kari-S population were poorly distributed into different frequency classes (Figure 6), showing the presence of a large proportion of alleles of higher frequency classes at several points. In contrast, the frequency distribution of Kari-M and Kari-L populations followed a normal L-shape graph showing a high proportion of alleles in low-frequency classes (0.01–0.20).

## 4. Discussion

### 4.1. Genetics of Gestation Length Variation in Sheep

Variation in sheep’s GL has been reported in previous literature. Studies have shown that GL variation in sheep is breed specific [25] with high heritability and low environmental influence [26]. Significantly shorter GL (137.1 ± 0.81 days) has been reported for Nigerian Balami sheep compared to other Nigerian sheep breeds (150.3 ± 0.61 to 153.3 ± 0.60 days) [27]. Similarly, South African Dorper sheep have been shown to have a consistently shorter GL (146.5 days) compared to other sheep breeds (151.6 days) when maintained together [28]. These studies suggest a genetic basis for GL variation in sheep. In the contrast to the previous study, in the current study, we have observed higher variation in GL among individuals belonging to one sheep breed, these results may explain specific DNA regions responsible for GL variation in sheep. Although the sheep in the current study were kept in their natural habitat, experimental procedures, i.e., estrous synchronization and ultrasound examinations may also induce variation in GL as previously suggested [29].

In general, microsatellites are found in the genomic regions where the mutation rate is high which could impair gene expression, therefore it is said that their prevalence is lower in gene areas [30]. Recent studies in fish indicate the dominance of microsatellite markers in the non-coding region of the genome, however, markers with mono-, di-, and tri-nucleotide repeats are more common in the coding region [31]. Other studies have suggested their potential role in the regulation of DNA replication, transcription, and translation [32]. In the current study, most of the microsatellite markers revealed moderate genetic diversity among the three Kari sheep subtypes, with the Kari-S as a separate divergent group. Only two markers (MAF214 and ILSTS5) showed significant association with GL, suggesting their role in modifying the genes controlling GL in sheep. Both markers have shown polymorphism in sheep in previous studies and marker MAF214 is a dinucleotide repeat which may be linked with a coding region in the sheep’s genome [33,34]. However, these markers showed only moderate correlation values. Furthermore, in the subsequent tests, the markers were found neutral for selection. Therefore, an in depth analysis of these markers is required in experimental flocks before incorporation in selection programs.

### 4.2. Genetic Diversity among Subtypes

The markers used in the recent study are highly explicit ovine-specific microsatellite markers recommended by the FAO [35]. Genetic markers are usually classified based on their PIC values, if the PIC value of a marker is less than 0.25, it is considered to be less informative while those whose PIC value is greater than 0.5 is considered to be highly informative for quantitative genetic studies [19]. The PIC values between 0.25 and 0.5 are considered to be convincingly informative. In the current study, the mean PIC values calculated in the three subtypes of Kari sheep were highly informative and were in accordance with the Hazaragie sheep (0.534) of Afghanistan [36]. One marker in Kari-L (OarFCB193) could be considered less informative having PIC values less than 0.25. Another microsatellite marker (BM1824) has been found less informative previously in the Michni sheep population of Pakistan [37].

The lowest Na was observed in the Kari-S subtype, suggesting the loss of respective alleles and genetic diversity probably due to its isolated habitat or small size. Comparatively higher Na in Kari-M and Kari-L suggest that their genetic diversity has been maintained, attributed to their comparable size and wide distribution in the valley (field observations). The Na in Kari-M and Kari-L populations were in accordance with Awassi sheep from Jordan and lower than Arabian sheep breeds [38]. Mean Ne and AR were less than Na in all the Kari subtypes, suggesting the predominance of certain alleles across the polymorphic loci in each of the subtypes [39]. Kari-L and Kari-M shared a maximum number of alleles (57.6%). These two populations were also genetically close to each other as evident from the genetic distance between the populations. This may be because of the geographical distribution of the Kari subtypes, and the high altitude difference between different regions of Chitral [2]. In our previous studies, three Pakistani sheep breeds Hashtnagri, Balkhi, and Michni populations found in the adjacent breeding region (in the central valley of KP) have been found to share more alleles (75–86%) [40].

The Kari-S population can be distinctly characterized based on their higher number of unique alleles. Of the total unique alleles found in Kari-S in the current study, the frequency of five alleles exceeded 0.2. High-frequency unique alleles were also found in the other three subtypes of Kari in the current study. These Unique alleles may serve as potential genetic markers for shorter gestation lengths in sheep. A higher number of unique alleles (12) have been found previously in the Pakistani Michni sheep population using the same microsatellite markers [40]. The results of the Ewens-Watterson test showed that most of the loci in all three subtypes were neutral for selection, suggesting the absence of a selection strategy or selective sweep [41]. Few of the microsatellite loci in Kari-S (two) and Kari-L (three) subtypes, in the current study, could be identified as the regions affected by selection. These regions have been considered as a tool for the identification of specific differences between subtypes [42]. Markers MAF214 and ILSTS which showed a correlation with GL in the current study were found neutral for selection. These results suggest that variations in these markers are random and not under external selection pressure. Thus overall the marker may be neutral for selection but some alleles may have an association with the trait.

### 4.3. Assigning Individuals to Subtypes

The principle of assigning an individual to a respective population is based on the probability of a genotype belonging to a sampled reference population [21]. In the current study, different methods showed a different percentage of correctly assigned individuals to their respective subtypes. The reliability of these methods depends on the conditions of the study. For example, Cornuet et al. [43] have reported that if a population is constantly changing in terms of mutations at a markers site then the Bayesian method is thought to be superior to the other methods. Contrary to this if a population deviates from HWE & linkage disequilibrium, then the distance-based method is preferred over the Bayesian method. Similarly, if a population is genetically more diverse then the distances based method is preferred [44].

Analyzing the situation of the present study, the genetic diversity of the Kari subtypes was comparatively less than previously reported for other sheep breeds. Furthermore, most of these subtypes were found in HWE. These results indicate the suitability of the Bayesian and frequency-based methods, which showed a higher precision for the individuals belonging to the Kari-L subtype, indicating Kari-L as the primitive breed from which Kari-S and Kari-M originated. In our previous study, another Pakistani sheep breed (Michni) has been reported to have a higher assignment accuracy (82–98%); whereas, Hashtnagri sheep has been found to have less accuracy (2–88%) [40] compared to the current populations. The STRUCTURE results showed a higher precision of individuals belonging to Kari-S suggesting strong genetic differentiation of this subtype. The genotypes (individuals) of the Kari-M and Kari-L subtypes were assigned jointly to two groups, indicating that these two subtypes share an admixed ancestry [23]. This was further supported by the lower genetic distance between Kari-M and Kari-L subtypes.

### 4.4. Inter-Subtype Breeding and Differentiation

The pairwise FST value equal to 0.05 suggests moderate differentiation among the populations [45]. Further lower values of FST indicate a low level of differentiation. In the current study, FST values among the pairs of the three Kari subtypes were higher than 0.05, indicating a high level of genetic differentiation between the subtypes. The higher FST values between the Kari-S and the other Kari subtypes were in accordance with its genetic distinctness identified in the STRUCTURE analysis in this study.

Total inbreeding (FIT) and the gene flow (NM) between the subtypes in this study were lower compared to the previous studies on other Pakistani sheep breeds, Balkhi, Hashtnagri, and Michni [40]. The FIT and NM were higher between Kari-M and Kari-L subtypes compared to other combinations. Having a similar phenotype and geographical distribution, the possibility of inbreeding between these two subtypes could not be excluded. In the current study, Kari-M and Kari-L subtypes also showed less genetic distance and more ancestral admixture which may be because of the higher level of inbreeding and gene flow between these two subtypes.

### 4.5. Population Bottleneck and Mutation Drift

As a population size reduces, its number of alleles reduces faster than the heterozygosity at the given loci, resulting in higher heterozygosity than expected and causing a genetic bottleneck [24]. If a population can maintain its allele number, it is said to be in mutation drift equilibrium. Bottleneck software provides the opportunity of computing Mutation drift equilibrium using the allele frequency data under different mutation models. In Step-wise Mutation Model (SMM) the changes in fragment length (increase or decrease in allele size) are exploited; the Two-Phase Model (TPM) considers substitution mutation having no change in allele size; while the Infinite Allele Model (IAM) (Crow and Kimura Model) considers heterozygosity as a function of consistent changes in allele frequency in a population under the consistent pressure of mutation and subsequent elimination dominated by mutation drift. For microsatellite data, the results obtained under SMM are more acceptable compared to the other two mutation models [46].

The software evaluates these mutation models using different statistical tests (see Materials and Methods). A similar number of loci with heterozygosity excess as expected in the Kari-M subtype suggest the absence of a genetic bottleneck in the Kari-M population, indicating that this population has been able to maintain its population size in the recent past. The Kari-L subtype also showed a lack of genetic bottleneck indicated by a normal L-shape graph. The bottleneck analysis of the Kari-S subtype shows the presence of a small number of individuals, which may require conservation for its unique genetic value for GL and future breed improvement programs.

Animal biodiversity has a significant role in maintaining environmental health and any losses in biodiversity will have a direct or indirect adverse effect on the well-being of human [47]. Sheep raising is an important source of livelihood in the hilly terrain of Pakistan, where Kari did evolve over the centuries as a mini sheep in the Chitral valley between steep mountains with scarce and seasonal availability of feed. Besides these unique reproductive traits, Kari proved to be a vibrant breed thriving under the subtropical climate of central KP, and has sufficient resistance to the prevalent and emerging diseases. This sheep resource is a pleasant addition in the sheep diversity and can be used as a biological asset to introduce accelerated lambing and enhanced heat and disease resistance, which will ultimately benefit humans. This interrelationship between human, animal and environmental health gave rise to the “One Health” concept—the joint interdisciplinary effort to achieve optimal health for people, animals, and the environment [48].

## 5. Conclusions

Kari is one of the thin-tailed sheep breeds of the Khyber Pakhtunkhwa that can be divided into three subtypes based on differences in gestation length. These subtypes are also diverse genetically and each possesses unique alleles that could serve as genetic markers. The unique and associated alleles at different microsatellite markers require further analysis at the sequencing level and their proximity or linkage with other genomic regions. After evaluation in large population sizes and deciphering their molecular mechanism microsatellite alleles identified in this study may be developed as genetic markers for shorter gestation length in sheep. The data suggest further study of these breeds for variation in their specific characteristics such as productivity and reproduction, and exploitation of their unique values for improving sheep reproductive performance in terms of gestation length. Apart from the shorter gestation length, Kari-S was also the most distant of the three subtypes with a unique genetic structure. Other genomic regions in the Kari-S subtype may also be explored to understand the genetic basis for shorter gestation length in sheep.

## Figures and Tables

**Figure 1 animals-12-03292-f001:**
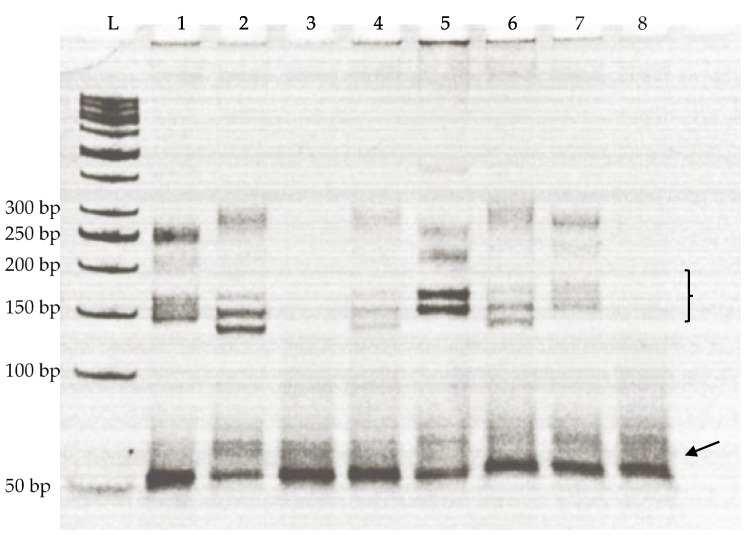
A representative image of polyacrylamide gel electrophoresis showing microsatellite banding pattern in eight individuals of Kari sheep. The sizes of the PCR product bands in samples (1–8) were calculated from the 50 bp ladder (L). Arrow indicates the primer dimmers. Bracket indicates the bands that came in the size range of the given microsatellite marker. Different size bands within the range were considered as different alleles for the marker. The non-specific bands above the expected region of the gel were not included in the analysis.

**Figure 2 animals-12-03292-f002:**
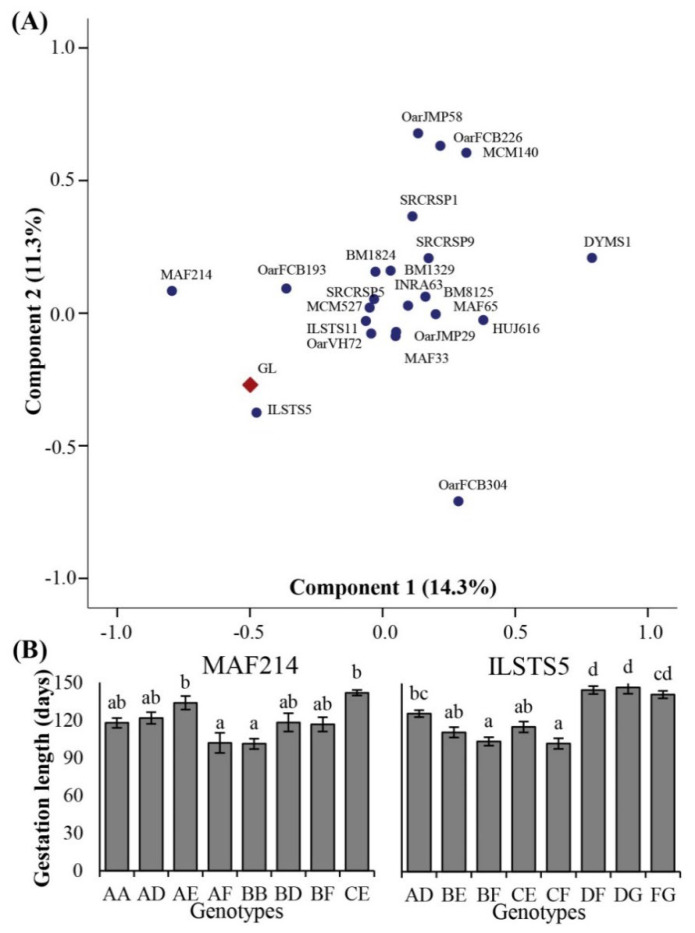
Association analysis between microsatellite loci and gestation length in Kari sheep. (**A**) Principal component plot showing clustering of closely associated parameters. (**B**) Mean gestation length among different genotypes of highly correlated loci. Labels on the bars (letters a–d) show a comparison among genotypes (*p* < 0.05).

**Figure 3 animals-12-03292-f003:**
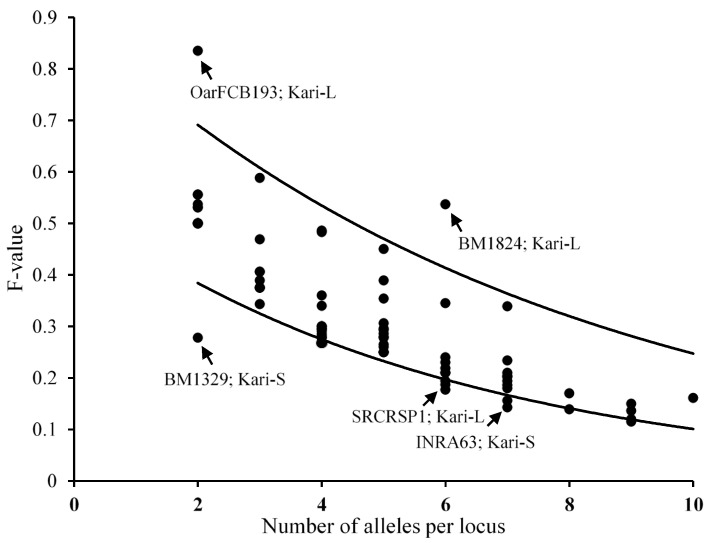
Ewens-Watterson test of neutrality of markers in the three Kari subtypes. The lines represent the upper and lower 95% confidence intervals. The dots show the loci. Arrows showing the markers (labeled) falling outside of the confidence intervals.

**Figure 4 animals-12-03292-f004:**
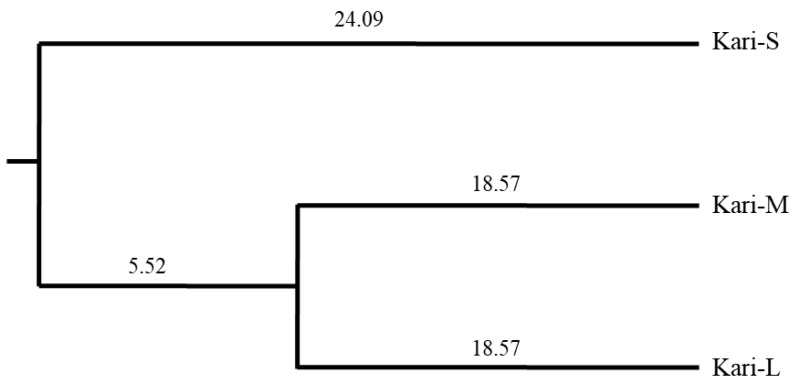
Dendrogram based on Nei’s unbiased genetic distance using the neighbor-joining method.

**Figure 5 animals-12-03292-f005:**
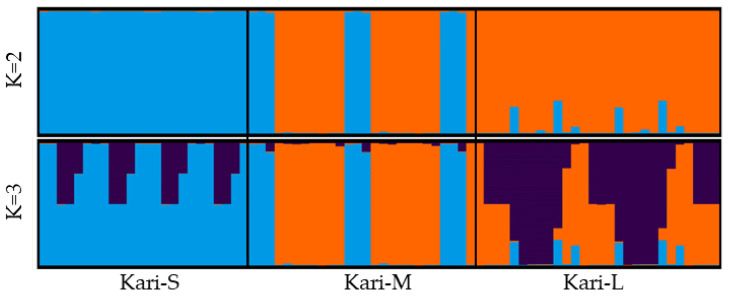
Summary plot showing the membership probability of individuals in the predefined subtype at different K values. Each bar represents an individual. Note that at K = 3 a cluster made by Kari-M showed a proportion of Kari-S in the ancestry; while an ancestral admixture of Kari-L was observed in the group made by Kari-S.

**Figure 6 animals-12-03292-f006:**
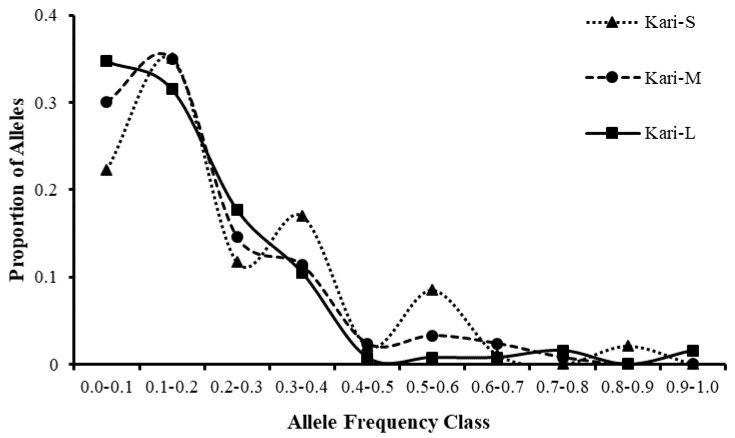
Allele frequency distribution graph of the three Kari subtypes. A mode shift can be seen in Kari-S subtype, while Kari-M and Kari-L subtypes followed a normal L-shape graph.

**Table 1 animals-12-03292-t001:** Variation in gestation length among the three Kari subtypes and its association with litter size and birth weight.

Subtype	Sample Size	Gestation Length	Litter Size	Birth Weight
Kari-S	24	100.7 ± 1.8 ^a^	1.54 ± 0.12	1.92 ± 0.13
Kari-M	26	123.1 ± 1.0 ^b^	1.57 ± 0.12	2.23 ± 0.12
Kari-L	28	143.8 ± 1.5 ^c^	1.53 ± 0.10	2.26 ± 0.11
Sig.		***	ns	ns

Key: *** = *p* < 0.001; ns = non-significant. Superscripts letters (a–c) show difference within the column (*p* < 0.05).

**Table 2 animals-12-03292-t002:** Mean values of gene diversity overall loci in different sheep populations.

Subtype	Np	N_T_	Na	Ne	A_R_	Ho	He	F_IS_	PIC	HWE (*p* Value)
Kari-S	21	92	4.318	3.451	2.790	0.812	0.659	−0.136	0.578	0.275
Kari-M	22	121	5.454	3.798	2.822	0.761	0.722	0.006	0.628	0.098
Kari-L	21	119	5.454	4.171	2.889	0.757	0.691	0.040	0.622	0.152

Key: A_R_, allelic richness; F_IS_, within-population inbreeding coefficient; He, expected heterozygosity; Ho, observed heterozygosity; HWE, Hardy-Weinberg equilibrium; Na, the observed number of alleles; Ne, expected number of alleles; Np, number of polymorphic loci; N_T_, the total number of alleles.

**Table 3 animals-12-03292-t003:** The average probability of individuals correctly assigned to their respective subtype using different statistical criteria.

Subtype	Frequency Criteria	Bayesian Criteria		Distance Criteria				
		R&M	B&L	D_S_	D_M_	D_A_	D_C_	D_AS_
Kari-S	66.67	66.67	50.00	83.33	83.33	66.67	66.67	83.33
Kari-M	63.64	54.55	72.73	72.73	72.73	72.73	72.73	54.55
Kari-L	83.33	83.33	75.00	75.00	66.67	83.33	83.33	58.33

Key: R&M, Ranala and Mountain; B&L, Baudouin and Lebrun statistics; D_S_, Nei’s standard distance; D_m_, Nei’s minimum distance; D_A_, Nei’s average distance; D_c_, cord distance; D_As_, shared allele distance.

**Table 4 animals-12-03292-t004:** Pairwise genetic association of the three Kari subtypes based on microsatellite data.

	Kari Subtypes	Kari Subtypes	
		Kari-M	Kari-L
Population differentiation (F_ST_)	Kari-S	0.147 ± 0.021 *	0.149 ± 0.022 **
	Kari-M		0.105 ± 0.022 *
Total inbreeding (F_IT_)	Kari-S	0.030 ± 0.047 *	0.015 ± 0.042
	Kari-M		0.046 ± 0.041 **
Gene flow (N_M_)	Kari-S	3.779 ± 0.753	4.234 ± 0.721
	Kari-M		9.315 ± 2.609
Genetic distance	Kari-S	0.285	0.280
	Kari-M		0.205

Key: *, *p* < 0.05; **, *p* < 0.01 under exact-G test assuming random matting within the samples.

**Table 5 animals-12-03292-t005:** Values for mutation drift equilibrium in Kari subtypes under different mutation models and statistical tests.

Subtype	Mutation Model	Sign Test			Standardized Differences Test		WS Rank Test
		Hee	He	*p*	T2	*p*	*p*
**Kari-S**	IAM	12.46	19	0.003	5.07	0.000	0.000
	SMM	12.74	17	0.048	3.06	0.001	0.008
	TPM	12.58	19	0.003	4.16	0.000	0.000
**Kari-M**	IAM	13.47	19	0.013	3.91	0.000	0.000
	SMM	13.59	16	0.210 *	0.82	0.205 *	0.172 *
	TPM	13.48	17	0.098 *	2.63	0.004	0.009
**Kari-L**	IAM	13.11	20	0.002	4.19	0.000	0.000
	SMM	13.24	18	0.032	1.70	0.044	0.007
	TPM	13.33	18	0.035	2.99	0.001	0.000

Key: He, number of loci with heterozygosity excess; Hee, expected number of loci with heterozygosity excess; IAM, infinite allele model; *p*, probability value for heterozygosity excess; SMM, stepwise mutation model; TPM, two-phase model; T2, test 2; WS, Wilcoxon-Sign; *, *p*-values indicating non-significant differences among He and Hee.

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
