# Peer review of "Microsatellite Analysis Revealed Potential DNA Markers for Gestation Length and Sub-Population Diversity in Kari Sheep"

_animals, 2022, doi:10.3390/ani12233292_

Round 1

Reviewer 1 Report

Dear Authors

The work is very interesting but need a great improuvement. I have an intersting idea to improuve the work, you can contact me.

Best regards

Author Response

Dear reviewer 1,

We are thankful for the constructive revision and a number of gaps identified by the reviewer in the manuscript. Following is a point-by-point reply to your comments and the manuscript revised accordingly.

Regards,

Comments are in plain text, author responses are bold and italic.

Title change required.

Title has now been updated as per the reviewer comment with the word “characterization” as suggested by the reviewer replaced with “diversity”.

Formatting and language corrections required.

Author response: the rearrangements and formatting suggested by the reviewer have now been corrected in the manuscript.

You need add a part of gene map analysis

Author response: The microsatellite markers used in this study are scattered throughout the genome with one or two markers per chromosome. We have tried to perform a gene map analysis; however, with such a low density of markers the maps do not contribute much to the results. Instead we have now added the chromosomal location and position of markers at each chromosome in Table S1.

I don’t find the result of Geneclass in the result part.

Author response: The Geneclass results have been presented in section 3.4 and Table 3 in the revised manuscript.

Structure result need to be corrected by clump software

Author response: The structure results have now been corrected by CLAMAK software.

Follow the same plan you write in materials and methods

Author response: The presentation of different content has now been synchronized in Materials and Methods, Results, and Discussion.

The Ewens Watterson test result is in contradiction about your PCA result where you found a strong correlation between GL and MAF and ILSTS microsatellites.

Author response: The Ewens Watterson analysis tests the overall neutrality of markers for selection. The results suggest though these markers were neutral for selection, but it does not exclude the possibility of having unique alleles that show association with gestation length. This has now been elaborated in the results and discussion sections.

Where is the part of genetic structure and diversity in conclusions?

Author response: Parts of the genetic diversity and structure have now been added in conclusions section.

Reviewer 2 Report

The study is really interesting examining the genetic profile of a local Pakistani sheep breed, with a very interesting trait, a shorter gestation period. Although the genetic structure as revealed by the studied microsatellites is very well statistically analysed and discussed, the suggestion that MAF214 and ILSTS5 markers are associated with gestation period is not well supported and discussed. For this purpose a very extensive literature review should have been added in the discussion for previous associations of these two loci with gestation or other traits in other sheep breeds or populations. Here there is only half a paragraph mentioning that they are polymorphic (lines 354-355). Moreover the sample size is not very big for such an important association. I am afraid that as it is I would not recommend publication, instead if this part (that assocaition) would be deleted.

Other comments

line 16 and elsewhere: The word “race” is not the most correct term. I think it should be replaced with “breed” here and in the entire manuscript

The way the abstract presents the results and the conclusions is okay, i.e. that Kari-S is a genetically distinct subtype with higher genetic differentiation and distance from the other two populations. However I disagree with the sample summary at the point mentioning that two microsatellites may serve as genetic markers to identify short gestating ewes. This statement is very straightforward. Microsatellite markers in farm animals such as sheep have been studied a lot and an association with a productive trait like the gestation period is very important. In such a case more info for these two markers has to be provides: have they be linked with other traits previously? Are they not selectively neutral? Are they located within an intron maybe?

In line 47 “In most cases sheep are raised for meat”, does this refer to Pakistan? If so please write it. If this refers to global, it is not correct.

2.4. I am not totally sure how the 50 bp ladder was sufficient to score the alleles in a 10% polyacrylamide gel. Please explain better, probably provide the allele numbers. Where they all trinucleotides? Dinucleotide motifs (repeat length) would be hard to be read

2.5 Did you run any statistical analysis for allelic dropouts and other potential artefacts determination?

The test for neutrality is not in agreement with the two markers that were associated with the gestation length (MAF214 and ILSTS5). Since these two markers are associated with this trait how is it possible to be indicated as neutral?

Author Response

Dear Reviewer 2,

Thanks for the comments and constructive revisions of the manuscripts. We have now updated the manuscript accordingly. Following is a point-by-point reply to your comments on the manuscript.

Regards,

Comments are in plain text, author responses are bold and italic.

The study is really interesting examining the genetic profile of a local Pakistani sheep breed, with a very interesting trait, a shorter gestation period. Although the genetic structure as revealed by the studied microsatellites is very well statistically analysed and discussed, the suggestion that MAF214 and ILSTS5 markers are associated with gestation period is not well supported and discussed. For this purpose a very extensive literature review should have been added in the discussion for previous associations of these two loci with gestation or other traits in other sheep breeds or populations. Here there is only half a paragraph mentioning that they are polymorphic (lines 354-355). Moreover the sample size is not very big for such an important association. I am afraid that as it is I would not recommend publication, instead if this part (that assocaition) would be deleted.

Author response: Further discussion points have now been added to the use of markers MAF214 and ILSTS5 in the discussion section lines 377-380, lines 418-422. These markers showed low correlation values and found neutral for selection, thus may not be appropriate to be used as genetic markers. This has now been clarified in the text that the validation of these associated markers requires further analysis in large populations. We were unable to find recent studies on genetics of gestation length variation in sheep. These markers have been extensively used for genetic diversity estimation, but no study was found for trait associations using these markers. The authors reason that this analysis would be of interest to the readers and may form a basis for the use of microsatellite markers for trait association in sheep.

Other comments

line 16 and elsewhere: The word “race” is not the most correct term. I think it should be replaced with “breed” here and in the entire manuscript.

Author response: The word “race” has now been replaced with “breed”. The “race” was only mentioned in the simple summary to address general audience who may not be familiar with breed. However, now it has been changed as per the reviewer’s suggestion.

The way the abstract presents the results and the conclusions is okay, i.e. that Kari-S is a genetically distinct subtype with higher genetic differentiation and distance from the other two populations. However I disagree with the sample summary at the point mentioning that two microsatellites may serve as genetic markers to identify short gestating ewes. This statement is very straightforward. Microsatellite markers in farm animals such as sheep have been studied a lot and an association with a productive trait like the gestation period is very important. In such a case more info for these two markers has to be provides: have they be linked with other traits previously? Are they not selectively neutral? Are they located within an intron maybe?

Author response: the statement that these markers can be used to identify short gestating ewes has now been rephrased in simple summary.

In line 47 “In most cases sheep are raised for meat”, does this refer to Pakistan? If so please write it. If this refers to global, it is not correct.

Author response: This whole paragraph has now been removed from the manuscript as per the comment of reviewer 3.

2.4. I am not totally sure how the 50 bp ladder was sufficient to score the alleles in a 10% polyacrylamide gel. Please explain better, probably provide the allele numbers. Where they all trinucleotides? Dinucleotide motifs (repeat length) would be hard to be read

Author response: the 50 bp ladder provided enough gradation to identify small differences in band sizes. This has now been explained in section 2.4. Furthermore, a representative polyacrylamide gel image with scoring method explained has now been added as Figure 1 in the revised manuscript. The band sizes are given in Table S3. We agree that it would be difficult to identify a two bp difference in this system; however, in most cases there would be multiple dinucleotide repeats differences among bands, which further increase the band size differences. Different allele sizes identified through this approach are presented in Table S3.

2.5 Did you run any statistical analysis for allelic dropouts and other potential artefacts determination?

Author response: Analysis to calculate the frequency of Null alleles has been performed using genepop software. The results have been presented in Table S3 in the revised version of supplementary tables.

The test for neutrality is not in agreement with the two markers that were associated with the gestation length (MAF214 and ILSTS5). Since these two markers are associated with this trait how is it possible to be indicated as neutral?

Author response: The Ewens Watterson analysis tests the overall neutrality of markers for selection. The results suggest though these markers were neutral for selection, but it does not exclude the possibility of having unique alleles that show association with gestation length. Furthermore, the correlation values of the two markers with the trait were also low. This has now been elaborated in the results and discussion sections.

Reviewer 3 Report

Introduction.

The first paragraph is totally redundant.

The objectives of the study must be presented clearly.

Procedures

Please describe the protocol for reproductive control with details.

Please describe the primers for PCR

Results

Table 3. Please transfer to supplementary material.

Discussion

The discussion must be divided into sub-sections for easier flow of reading.

Also, the authors must add some references presenting similar studies in other sheep breeds and must compare the findings to their own.

Author Response

Dear Reviewer 3,

The authors are thankful for the constructive revisions and points raised that help in improving the manuscript. Following is a point-by-point reply to your comments and manuscript revised accordingly.

Regards,

Comments are in plain text, author responses are bold and italic.

Introduction.

The first paragraph is totally redundant.

Author response: The first paragraph has now been removed from the introduction.

The objectives of the study must be presented clearly.

Author response: Three clear objectives of the study have now been described at the end of the introduction section.

Procedures

Please describe the protocol for reproductive control with details.

Author response: Reproductive controls of the study including confirming open ewes, controlled mattings and Author response: subsequent confirmation of successful conception have now been added to Materials and methods.

Please describe the primers for PCR.

Author response: List primers have now been included as Table S1.

Results

Table 3. Please transfer to supplementary material.

Author response: Table 3 has now been transferred to supplementary material as Table S2. The numbering for rest of the tables in the main text have now been updated accordingly in the revised manuscript.

Discussion

The discussion must be divided into sub-sections for easier flow of reading.

Author response: subsections have now been made for the discussion.

Also, the authors must add some references presenting similar studies in other sheep breeds and must compare the findings to their own.

Author response: The available literature on gestation length variation in sheep has now been cited in discussion, with comparisons to our findings.

Round 2

Reviewer 2 Report

After the responses of the the authors, I believe the paper my be published in the current form

Author Response

Dear reviewer,

Thanks for the recommendation of adding extra content related to the significance of the study. We have now updated the manuscript as per the given comment. Following is the author’s response to your comment.

Regards,

Reviewer comments are in plain text, author’s responses are in bold and italic.

The manuscript has been improved.

Before acceptance, the authors must add a new paragraph in the discussion to underline the clinical significance of their work and the importance for sheep heath management within the one-health concept.

Response: A new paragraph has now been added to the discussion as was suggested by the reviewer.

Reviewer 3 Report

The manuscript has been improved.

Before acceptance, the authors must add a new paragraph in the discussion to underline the clinical significance of their work and the importance for sheep heath management within the one-health concept.

Author Response

(The authors gave the same response as above.)
